# Volcanism and long-term seismicity controlled by plume-induced plate thinning

**Raffaele Bonadio** [1] ✉, **Sergei Lebedev** [1,2], **David Chew** [3], **Yihe Xu**[1,4], **Javier Fullea**[2,5] **& Thomas Meier**[6]

Mantle plumes, the hot upwellings from the Earth's core-mantle boundary, are thought to trigger surface uplift and the emplacement of large igneous provinces (LIPs). Magmatic centres of many LIPs are scattered over thousands of kilometres. This has been attributed to lateral flow of plume material into thin-lithosphere areas, but evidence for such flow is scarce. Here, we use abundant seismic data and recently developed methods of seismic thermography to map previously unknown plate-thickness variations in the Britain-Ireland part of the North Atlantic Igneous Province, linked to the Iceland Plume. The locations of the ~ 60 Myr old uplift and magmatism are systematically where the lithosphere is anomalously thin at present. The dramatic correlation indicates that the hot Iceland Plume material reached this region and eroded its lithosphere, with the thin lithosphere, hot asthenosphere and its decompression melting causing the uplift and magmatism. We demonstrate, further, that the unevenly distributed current intraplate seismicity in Britain and Ireland is also localised in the thin-lithosphere areas and along lithosphere-thickness contrasts. The deep-mantle plume has created not only a pattern of thin-lithosphere areas and scattered magmatic centres but, also, lasting mechanical heterogeneity of the lithosphere that controls long-term distributions of deformation, earthquakes and seismic hazard.

The North Atlantic Igneous Province (NAIP) was emplaced, starting at ~62 Ma, across a broad area from the Baffin Island and western Greenland in the west to Britain and Ireland in the east[1–5]. The simultaneous, voluminous volcanism across the province was characterised by high $^3$He/$^4$He ratios[6–8] and is commonly attributed to the Iceland Plume[5,9–12]. The broad scatter of the magmatic centres, however, is puzzling and fuels debate on whether multiple mantle upwellings or, alternatively, processes within the lithosphere and upper mantle unrelated to plumes may be responsible for the NAIP emplacement[4,13,14].

Widely scattered magmatic centres are common in other LIPs as well, including, for example, the Central Atlantic Magmatic Province[15],

the High-Arctic Large Igneous Province[16] and the currently active East Africa-Arabia magmatic province[17–19]. Lateral flow of hot plume material along thin-lithosphere channels has been proposed to explain the broad and uneven distribution of magmatism[5,10,11,17,19,20]. There is no direct evidence, however, for asthenospheric flow in LIPs that were emplaced tens or hundreds of million years before present, as the mantle flow today is different from that in the past.

The southeastern part of NAIP—the British and Irish Paleogene Igneous Province (BIPIP)—comprises volcanic centres and dyke swarms in northern Ireland, western Britain and the Irish Sea, including the voluminous Antrim Lava Group (Fig. 1, Supplementary Fig. 1)[2,8]. The volcanism was accompanied by magmatic underplating

[1]Department of Earth Sciences, Bullard Laboratories, University of Cambridge, Cambridge, UK. [2]School of Cosmic Physics, Dublin Institute for Advanced Studies, Dublin, Ireland. [3]Department of Geology, Museum Building, Trinity College Dublin, Dublin, Ireland. [4]School of Earth Sciences, Yunnan University, Kunming, China. [5]Department of Earth Sciences and Astrophysics, Universidad Complutense Madrid, Madrid, Spain. [6]Institute of Geosciences, Christian Albrecht University, Kiel, Germany. ✉e-mail: rb2075@cam.ac.uk

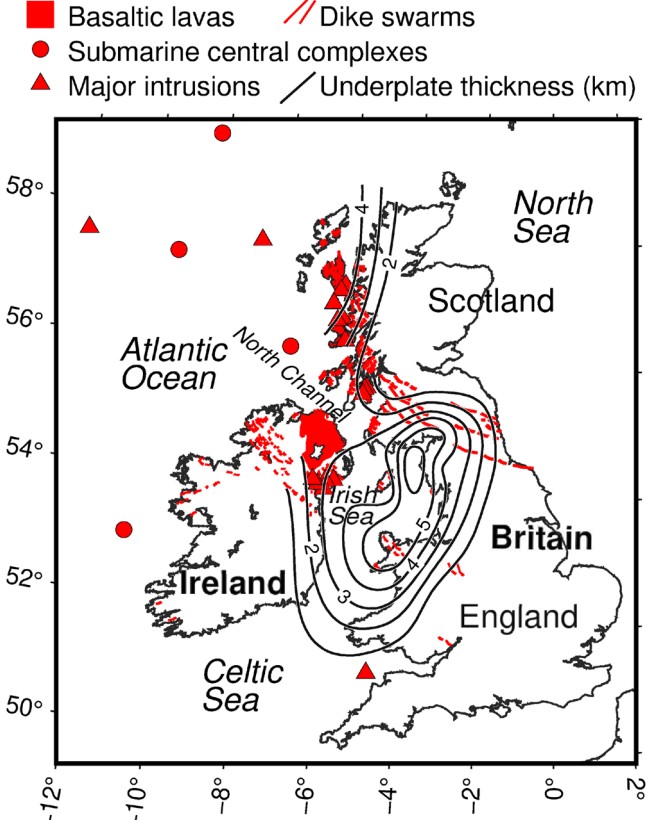

**Fig. 1 | The British and Irish Paleogene Igneous Province and the location of its major magmatic features[45,46].** Black contours indicate estimated magmatic underplating in km[21].

at the base of the crust beneath the entire area, as evidenced by controlled-source and receiver-function seismic data[21]. The emplacement of BIPIP at ~62–55 Ma was also accompanied by significant uplift and exhumation of crustal rocks—up to 3 km in total—as evidenced by low temperature thermochronology data[22-25] and sedimentation patterns[26,27]. The region experienced at least three distinct transient uplift events—around 62, 59, and 56 Ma. The Antrim Lava Group was emplaced during the first of these events, while the remainder of the NAIP emplacement and the final rifting that led to seafloor spreading are associated with the third and final uplift phase[28-31]. The uplift has been previously attributed to the magmatic underplating, which implied large underplated layer thicknesses of up to 8 km[12,21].

Plate reconstructions at 60 Ma show that BIPIP, when it was emplaced, was around a thousand kilometres away from the Iceland Hotspot, located in eastern Greenland at that time[5,30]. In order to connect BIPIP to the Iceland Plume, some models invoke a sheet-like upwelling, spanning from the hotspot to BIPIP[32,33] and others—lateral flow of the hot plume material[5,34]. As the paleo-flow in the asthenosphere cannot be observed today, there is no direct evidence supporting either of the models. Whether and how the Iceland Plume fed NAIP's scattered magmatic centres remains an outstanding question[4,5,13,35].

Here, we use abundant seismic data and recently developed tomographic and thermodynamic inversion methods to map in detail the thermal structure and thickness of the lithosphere in Britain and Ireland. The previously unknown lateral variations of the lithospheric thickness display striking correlations with the distributions of uplift, volcanism and underplating at ~60 Ma and, also, with the distribution of seismicity at present.

## Results
### Mapping the lithospheric thickness

Large parts of Britain have long been instrumented with broadband seismic stations, but Ireland had few until recently. Active and passive-source seismic studies mapped the crustal structure of much of the Isles[21,36], and traveltime tomography studies reported heterogeneity in the sub-lithospheric mantle[37,38]. The lithosphere, by contrast, was not mapped with seismic data and has normally been assumed, with few exceptions[39-41], to be of constant thickness. The deployment of the 20-station Ireland Array and the expansion of the Irish National Seismic Network to six stations in 2011 and 2012 created sufficient station coverage, and the accumulation of earthquake recordings over the following decade—sufficient data volumes for the detailed mapping of surface-wave velocities, the data type particularly sensitive to the thermal structure and thickness of the lithosphere, in Ireland and most of Britain. The data were recently used for Rayleigh-wave tomography using the optimal resolution tomography scheme[42], which solved the Backus-Gilbert problem[43] (see Methods) at every point using empirical model-error estimates and produced phase-velocity maps with known, laterally varying resolution.

Phase velocities of Rayleigh waves at 65 s period have primary sensitivity in the 65–130 km depth range[44]. Their lateral variations are a proxy for those of the depth of the lithosphere-asthenosphere boundary (LAB), if that occurs in an 80–120 km depth range. The 65 s phase-velocity map (Fig. 2a) hints at previously unknown variability of the lithospheric thickness, with the higher and lower phase velocities indicating thicker and thinner lithosphere, respectively. We compare the phase-velocity distribution with a compilation of the volcanic centres, dykes and magmatic underplating[21,45,46]. We also compare these data with the amounts of Palaeocene exhumation, obtained from our compilation of low-temperature thermochronology data (integrated apatite fission track and (U-Th)/He dating, often on vertical sampling profiles) and indicative of the net Palaeocene uplift[24,25].

The data show a remarkable correlation of the distribution of the Palaeocene uplift and magmatism and the variations of the Rayleigh-wave phase velocities (Fig. 2a). Both uplift and magmatism occurred predominantly in the circum-Irish Sea area where the 65 s, Rayleigh-wave phase velocity is relatively low, indicating relatively thin lithosphere. Furthermore, the amount of exhumation anti-correlates with the phase-velocity values, which shows that the uplift was the greatest where the lithosphere is the thinnest (Fig. 2b).

In order to investigate this relationship further, we determine the lithospheric thickness quantitatively. Although estimates of the lateral LAB-depth variations can be obtained from seismic velocity profiles and tomographic models[42], the seismic models are non-unique and very often imply implausible, oscillatory geotherms, which biases LAB-depth estimation[47]. A different approach is required to determine the relatively subtle variations in Britain's and Ireland's lithospheric thickness. Recently developed methods of seismic thermography invert seismic data directly for the temperature and thickness of the lithosphere, using computational petrology and thermodynamic databases and circumventing the non-uniqueness of intermediate shear-wave velocity models[47,48].

Both Rayleigh and Love surface wave measurements are required to determine lithospheric structure accurately. These two wave types are sensitive to the vertically and horizontally polarised S-wave speeds, respectively, which are different due to seismic anisotropy, caused by fabric within the rock at depth. Joint inversion of Rayleigh and Love dispersion data can resolve the anisotropy and isolate the isotropic average elastic properties related to temperature and composition.

We measured the Love-wave phase velocities between all possible pairs of stations in Britain and Ireland. A combination of inter-station cross-correlation and waveform inversion yielded interstation phase-velocity curves in broad period ranges, each averaged from many one-event curves[42]. We then used the optimal resolution, phase-velocity

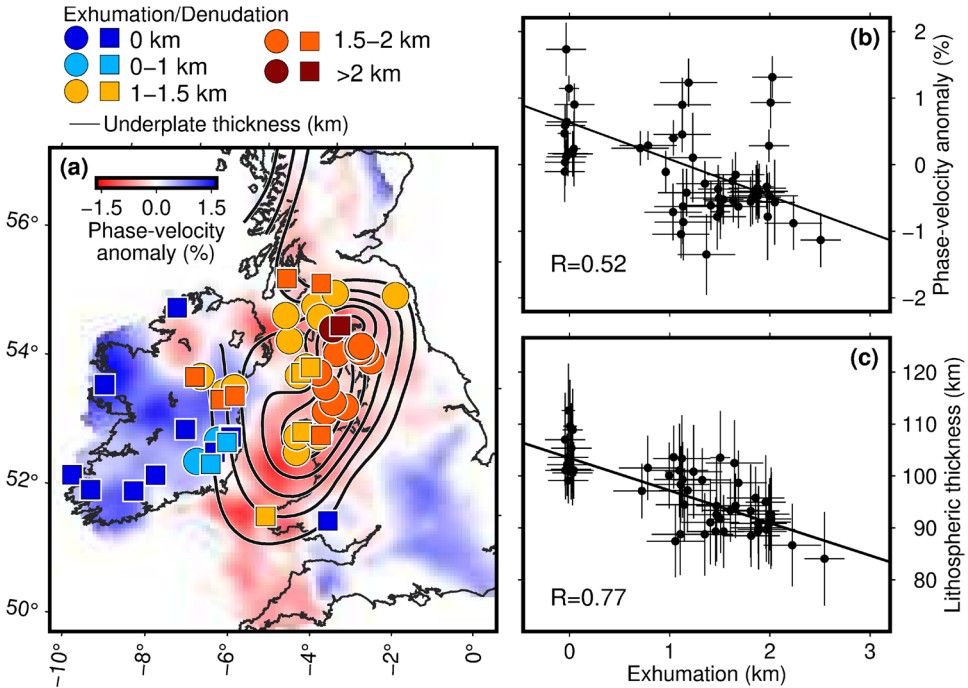

**Fig. 2 | The correlation of the Palaeocene exhumation[24,25] and the magmatism and underplating[21] (black contours as in Fig. 1) with Rayleigh-wave, phase-velocity anomalies at 65 s period and with the lithospheric thickness at the locations of exhumation measurements. a** The phase-velocity map for the 65 s Rayleigh wave. Exhumation estimates, indicative of the Palaeocene uplift, are over-plotted with coloured squares[24] and circles[25]. **b** Correlation of the phase-velocity anomaly and exhumation. The Pearson correlation coefficient is 0.52. **c** Correlation of the lithospheric thickness, yielded by seismic thermography, and exhumation. The Pearson correlation coefficient is 0.77.

tomography at densely spaced periods to obtain a set of Love-wave phase-velocity maps (Supplementary Fig. 2). Together with the already available Rayleigh-wave maps computed with the same methods[42], these yielded a pair of phase-velocity curves at every point, forming the input for the thermodynamic inversion.

Multi-observable thermodynamic inversions can integrate diverse data types with sensitivity to temperature and composition within the Earth[48]. Because seismic-velocity sensitivity to composition in the mantle is much smaller than to temperature, we can invert seismic data principally for temperature, with sensible assumptions on composition and other pertinent properties and with additional inversion parameters including anisotropy. Thus defined seismic thermography[47] focuses specifically on the temperature-seismic velocity relationship and aims to maximise the accuracy and completeness of the extraction of structural information from seismic data, which is essential for the accurate mapping of the thermal structure and thickness of the lithosphere[47].

The thickness of the lithosphere and temperature of the asthenosphere are the key parameters of the thermodynamic inversion, with additional parameters for radial anisotropy and the seismic-velocity structure of the crust, mantle transition zone and uppermost lower mantle. The forward problem is solved by computing the elastic parameters from temperature and composition at different depths[49], with synthetic phase velocities computed from those (see Methods).

Attenuation is calculated as a function of depth and temperature, based on mineralogical measurements and relationships[48,50]. The inverse problem is solved using a non-linear gradient search, with the forward problem solved directly at every step. The forward and inverse problems for each 1D profile are small, and the misfit distributions in the model space form structurally simple misfit valleys[47].

### Uplift and magmatism in thin-lithosphere areas

Thermodynamic inversions at selected locations (Fig. 3) illustrate the decrease of the lithospheric thickness from 100 to 110 km in central Ireland to 85–90 km in the circum-Irish Sea area. Computing the lithospheric thickness at every point where the exhumation data are available, we observe a strong correlation of the two datasets (Fig. 2c). No Palaeocene exhumation—and, thus, no uplift—are seen where the lithosphere is relatively thick. Progressively greater exhumation—and, thus, greater uplift—are where the lithosphere gets thinner.

The Palaeocene magmatic centres, intrusions and dike swarms map in the same thin-lithosphere area (Figs. 1 and 2a)[2,8,45,46]. The estimated lateral extent of magmatic underplating[21] also matches the circum-Irish Sea, thin-lithosphere area. The variations in the lithospheric thickness are relatively subtle and can only be mapped quantitatively with the thermodynamic inversion. The main pattern, however, is already seen in individual phase-velocity curves[42] and in phase-velocity maps (Fig. 2a), confirming that it is required by the data. The circum-Irish-Sea, thin-lithosphere area also matches, roughly, the location of late teleseismic arrivals reported earlier[37,38], which, however, could not be attributed to the lithospheric thickness variations because of the limited vertical resolution of teleseismic body waves.

The remarkable correlations show that the uplift and magmatism in Britain and Ireland 60 Myr ago were concentrated within a thin-lithosphere area, which was present beneath and around the Irish Sea. The lithosphere there at that time must have been thinner than it is at present, probably 60 km or less, sufficiently thin to enable extensive partial melting beneath it, feeding the emplacement of the Antrim Lava Group and other magmatic structures[2,51]. The thickening of the lithosphere since then—over the last ~60 Myr—was a 3D process that depended on the evolution of the sedimentary layer, thermal conductivity and radiogenic heat production in the sediments and the crystalline crust and the dynamics of the sublithospheric mantle. Our results yield a direct empirical estimate for the lithospheric thickening, which we infer to be around 30 km, from 55–60 km at 60 Ma to 85–90 km today.

The boundaries of the thin-lithosphere area observed have nothing in common with the boundaries of geological terranes, which

extend northeast-southwest from Scotland to Ireland (Fig. 4b). The onset of the uplift and volcanism at ~62 Ma indicates that the lithosphere was probably thinned at that time. The simultaneity of the basaltic volcanism in BIPIP and other locations scattered around the Iceland Hotspot suggests that the hot material brought up by the Iceland Plume spread laterally away from the hotspot in a number of channels, probably guided by pre-existing variations in the lithospheric thickness[5,10,11,17,19]. Felsic magmatism of NAIP age has also been dated recently at the far reaches of the province, as far as southern Britain and just west of Norway, also hinting at a possible relationship

with the Iceland Hotspot[52–54]. The lithospheric thickness between Iceland and Britain and Ireland is unknown, so there are no direct observations of a thin-lithosphere corridor that could guide the hot material towards the Irish Sea. Elongated low-velocity anomalies at 120–200 km depths, either side of the North Sea were reported in an earlier tomographic study and were interpreted as channels of hot asthenosphere[14], but they were too deep to reflect subtle variations in the LAB depth and may have stood out, instead, by contrast with the anomalously high velocities beneath the North Sea.

Our results present direct evidence for the lithospheric thinning beneath BIPIP and for the plate thinning to have controlled the distribution and emplacement of the NAIP intraplate basalts. The presence of a thin lithosphere and hot asthenosphere ponded beneath it can account for the surface uplift and the pervasive decompression melting in the shallow asthenosphere, resulting in the extensive magmatism[55,56]. The presence of numerous submarine magmatic complexes between Iceland and the Irish Sea (Fig. 1, Supplementary Fig. 1) gives an indirect indication for a thin-lithosphere corridor extending from Iceland to BIPIP, with the same plate thickness–magmatism relationship as we observe directly in the circum-Irish Sea area.

The thermodynamic inversions also resolve the temperature in the asthenosphere and show that its lateral variations beneath Britain and Ireland are small at present and do not correlate with the variations in the lithospheric thickness (Supplementary Fig. 3). This indicates that the hot asthenosphere that caused the lithospheric thinning, uplift and magmatism at 60 Ma has cooled, with no anomalously hot material ponded beneath the areas with thinner lithosphere at present. Substantial cooling of the hot asthenosphere had to occur already at the time of magmatism via the latent heat of melting[57,58], and the conductive cooling and 3D convection since that time have resulted in the weakly heterogeneous temperature distribution seen at present.

The spreading between Greenland and Eurasia starting at ~56 Ma[28] created a thin-lithosphere channel beneath the newly formed Mid-Atlantic Ridge. North Atlantic tomography[14,59] shows exceptionally low seismic velocities beneath Iceland and the adjacent portions of the Mid-Atlantic Ridge, which is where abundant magmatism has been localised in the Neogene[60]. The anomalies indicate hot, partially molten asthenosphere supplied by the Iceland Plume. Prior to the formation of the thin-lithosphere corridor beneath the ridge, this material must have spread laterally toward the different volcanic areas of NAIP.

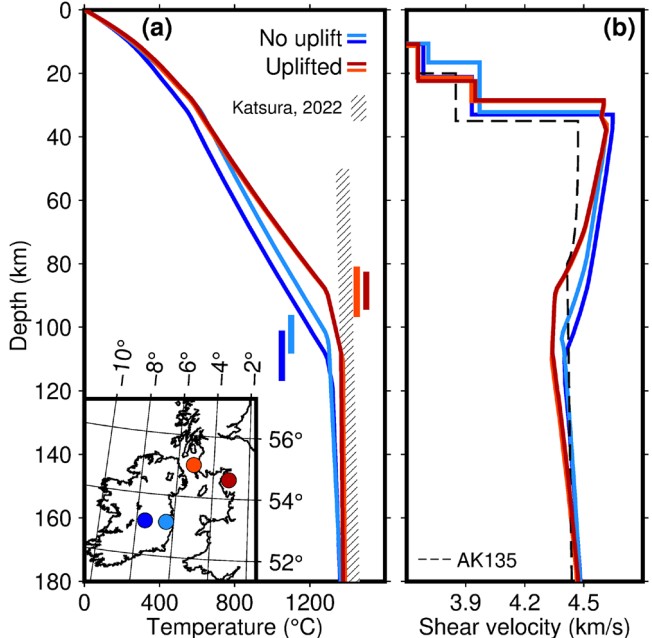

**Fig. 3 | Thermodynamic inversions at 4 example locations: two where Palaeocene uplift and magmatism took place (red) and two where they did not (blue). a** Geotherms and map of locations. Uncertainties on the lithospheric thickness are shown with the vertical bars. The lithospheric thicknesses in the models are 88.5 ± 5.9 km, 88.8 ± 7.9, 102.2 ± 6.0, and 108.9 ± 8.1. **b** Shear-wave velocity profiles. The average adiabatic temperature profile for the mantle[77] and the AK135 velocity model[99] are included in panels (**a**) and (**b**), respectively.

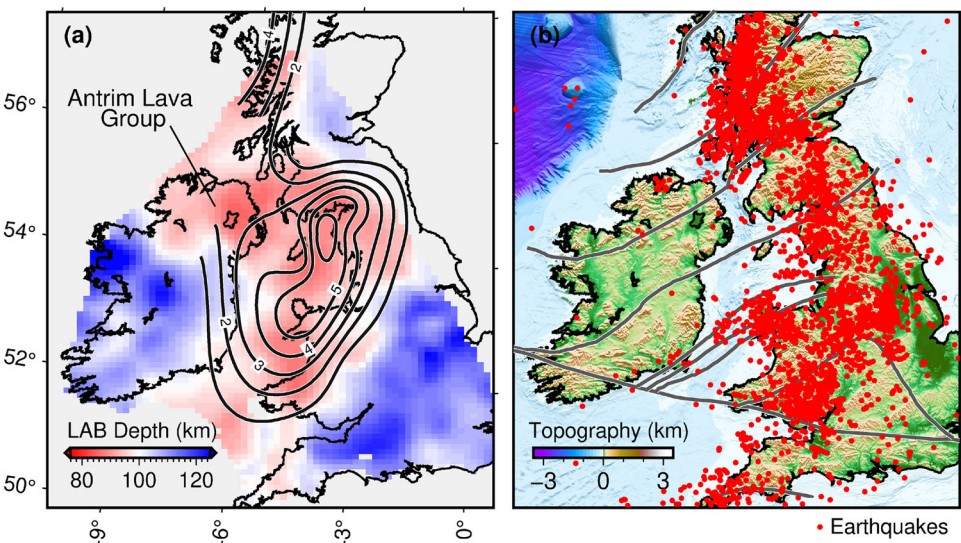

**Fig. 4 | Lithosphere-asthenosphere boundary (LAB) depth, magmatism and seismicity. a** LAB depth map yielded by seismic thermography. Magmatic underplating[21] and the Antrim Lava Group are located in an area of thin lithosphere. **b** Seismicity[63] and tectonic boundaries and major faults (grey lines).

The capture of the plume material by the ridge has cut off its supply to BIPIP and other locations of NAIP volcanism, at which point magmatism there waned and the lithosphere started cooling and thickening.

## Seismicity distribution controlled by plume-induced plate thinning

The intraplate seismicity in Britain and Ireland has a surprisingly uneven distribution and is poorly understood[61]. Most earthquakes are small but the largest possible earthquake in the UK is estimated at Mw 6.5[61]–larger, for example, than the devastating 2011 Christchurch, New Zealand, earthquake (Mw 6.2)[62]. It is, thus, important to understand the mechanism behind the seismicity distribution.

Based on Rayleigh-wave, phase-velocity maps, it has been proposed recently that seismicity in Britain and Ireland was localised in areas with thin lithosphere[63]. Using Love- and Rayleigh-wave tomography, along with our thermography methods, and the resulting quantitative maps of the thickness of the lithosphere, we can now determine how the lithospheric thickness shapes the distribution of earthquakes.

The first-order pattern we observe (Fig. 4) is that most earthquakes are concentrated in western Britain in the areas of relatively thin lithosphere and along thin-thick lithosphere boundaries. A large majority of Ireland's earthquakes are in its northernmost part[63], where the lithosphere is also thin. A thinner lithosphere is warmer (Fig. 3) and, thus, has lower mechanical strength and deforms more readily in response to regional tectonic stress. The seismogenic layer in the brittle upper crust is, typically, ~15 km thick, generally thinner in warmer compared to colder lithosphere but sufficiently thick in both cases to host small and intermediate earthquakes[64]. Greater deformation of weaker lithospheric blocks is accommodated by viscous flow in the deep lithosphere and numerous earthquakes (brittle failure) in the upper crust.

Crustal thickness variations could also affect the lithospheric strength but they are relatively small and do not correlate with seismicity in Britain and Ireland[42]. This confirms that the intraplate deformation and seismicity are localised by the variations in the thickness and temperature and, thus, the strength of the mantle lithosphere[65].

The distribution of earthquakes shows no relationship to tectonic boundaries or major faults (Fig. 4b). Instead, seismicity is concentrated in the same thin-lithosphere areas as the Palaeocene uplift and magmatism were and along lithosphere-thickness boundaries (Figs. 2 and 4). Our results reveal the manifold effects of the impingement of mantle plumes upon the lithosphere. The plumes carve the tectonic plates from below, and the thin-lithosphere regions they create focus the uplift and magmatism, which commence immediately and last until the supply of hot material to the area is terminated. By contrast, the lateral heterogeneity in the mechanical strength of the lithosphere created by the plumes endures for many tens of millions of years and controls long-term distributions of deformation, seismicity, and seismic hazard.

## Methods

The key methods used in this study are surface-wave tomography (which yielded the distributions of Love and Rayleigh wave phase velocities) and thermodynamic inversion (which extracted the information on the thermal structure of the lithosphere and asthenosphere from the phase velocities). Model-space projection was then used in order to estimate the LAB-depth uncertainty, and the LAB depth distribution was compared to the magmatism, exhumation and seismicity data.

### Optimal resolution, surface-wave tomography

We used the latest available data in the Ireland–Britain region in order to obtain numerous phase-velocity measurements in very broad frequency ranges, from periods as short as 4 s to those as long as 500 s. This enabled us to image the region with greater detail than previously available. Inter-station phase velocities at 11,238 station pairs that recorded simultaneously were obtained using a combination of an advanced recent implementation of the cross-correlation method[42,66,67] and multimode waveform inversion[68,69]. The phase-velocity measurements were then used to compute optimal-resolution phase-velocity maps[42] at a set of densely spaced periods. The optimal resolution tomography method solves the Backus-Gilbert problem, finding the shortest length scale over which the local average structure at every point can be determined with the variance under a specified amount[43]. In other words, the method determined the optimal resolving length (the width of the peak of the optimal averaging kernel), or optimal resolution, given the errors. This was achieved using empirical estimates of the phase-velocity model errors[42]. Love-wave phase-velocity maps, obtained in this study (Supplementary Fig. 2), were combined with the already available Rayleigh-wave maps[42] and provided a pair of phase-velocity curves at every point.

### Seismic thermography

The phase-velocity curves yielded by the phase-velocity maps form the input for the thermodynamic inversion[47,48,70]. The key parameters of the inversion include the thickness of the lithosphere and temperature within the asthenosphere between the LAB and 400 km depth. The lithosphere is the mechanically strong outer layer of the Earth, and it bottoms where the temperature becomes high enough for the convection to start. We adopt the 1300 °C temperature as this threshold and define the LAB by the 1300 °C isotherm[48,71–73]. The steady-state lithospheric geotherm is obtained by solving the 1D heat conduction equation with a temperature- and pressure-dependent mantle thermal conductivity[74,75]. The transition between the lithosphere and asthenosphere is parameterised with a boundary layer that has a variable, 5–50 km thickness and bottoms at 1400 °C. The heat transport within this layer is via both conduction and convection. Between the bottom of the buffer and the 410-km discontinuity, temperature is parameterised using three depth knots. As part of the inversion regularisation, deviations of temperature at these knots from the average adiabatic gradient of 0.5 K/km[76,77] are penalised. Other inversion parameters include seismic velocities in the crust, mantle transition zone and shallow lower mantle and radial anisotropy.

In the forward problem, we calculate the stable mineral assemblage assuming thermodynamic equilibrium and using the Gibbs free energy minimisation[49,78] and a thermodynamic database[79]. The bulk density and seismic velocities in the mantle are obtained from the density and elastic moduli of the constituent end-member minerals[74,80]. Average Phanerozoic upper mantle composition is assumed[81,82]. The chemical parameterisation assumes the major oxide system CFMAS (CaO–FeO–MgO–Al$_2$O$_3$–SiO$_2$), with the major oxides accommodated in the four primary upper mantle minerals: olivine, clinopyroxene, orthopyroxene and an Al-bearing phase (plagioclase, spinel or garnet, depending on pressure). Other, minor phases represent less than 5% of the total mineral assemblage. Attenuation (anelasticity) has a significant effect on the temperature–seismic velocity relationship and is calculated using a pressure- and temperature-dependent equation applied to the anharmonic velocities[48,82]. From the seismic velocities, density, radial anisotropy and attenuation, the synthetic phase-velocity curves are computed using the modes code MINEOS[83], modified for the travelling wave decomposition[84] and for phase-velocity computation speed[85,86].

The inverse problem is solved with a fully non-linear, Levenberg–Marquardt gradient search inversion algorithm. The forward problem is solved exactly at every step, with the synthetic phase velocities at each iteration computed starting with the temperature, pressure and composition distributions with depth. Regularisation is applied to penalise oscillatory radial anisotropy and asthenospheric temperature profiles, but does not constrain the lithospheric thickness.

The results of the inversions for selected four points summarised in Fig. 3 are presented in full in Supplementary Fig. 4. Surface elevation is computed based on local isostasy, integrating the density distribution from the surface to 400 km depth[48,70]. The models fit the observed elevation within less than 300 m, which corresponds to about 1% error in the bulk density of the crust[87], well below the uncertainty of our knowledge of it. The thermodynamic inversions of surface-wave data also predict the surface heatflow (Supplementary Figs. 3 and 4) and geothermal gradient, which makes them a useful tool for geothermal energy resource assessment at regional and global scales[88,89]. Errors of the phase-velocity curves are estimated by means of isolating their roughness that cannot be attributed to any Earth structure and is, thus, a measure of data noise[42].

## Model uncertainty

The model uncertainties on the LAB depth are estimated using the model-space projection approach[63,90] (Supplementary Fig. 5). Here, the misfit ellipsoid in the multi-dimensional parameter space is projected onto a one-dimensional sub-space of one parameter only, the LAB depth. For each location, a series of inversions is run while fixing the LAB depth at a value within a range around the best-fitting LAB depth, with a 1 km step. All the other parameters are allowed to vary freely, so that the trade-offs between them are fully accounted for in the model uncertainty estimate[91]. This procedure typically produces a parabolic function in the vicinity of the misfit minimum. Given a misfit threshold, that yields an estimate of the model error. In order to determine the threshold, we compute the difference between the misfit given by the best-fitting, preferred model and the misfit given by an unregularised inversion that overfits the data and is characterised by oscillatory profiles of anisotropy, temperature in the asthenosphere, and seismic velocities in the crust, transition zone and lower mantle. The unregularised inversion fits the noise in the data as much as possible, and the difference of its misfit with that corresponding to a best-fitting feasible model yields an estimate of the effect of noise on the model. We obtain the model-error threshold by adding this difference to the misfit given by the preferred model.

## Correlation between exhumation and lithospheric thickness

We compile a dataset of exhumation measurements from two published studies[24,25] that performed low-temperature thermo-cronological analyses and inverted the data to derive inverse thermal history models. Importantly, the combination of apatite fission track and (U-Th)/He data in each sample and, in many instances, the derivation of a thermal history from multiple samples within a pseudo-vertical profile, yields T-t histories with significantly higher resolution than legacy studies employing the modelling of apatite fission track of single samples alone. Phases of Paleogene cooling are interpreted as periods of exhumation and, by inference, uplift at key locations in the region. Uncertainties on the measurements are set depending on whether a given thermal history employed pseudo-vertical profile or single-sample estimates[24], or used averages of the minimum and maximum range given[25]. The uncertainty on the phase velocity is set according to the resolution length at each point, determined by the optimal resolution tomography[42]. The correlation between the LAB depths and exhumation measurements is computed using linear regression[92]. The Pearson coefficient is 0.52 and 0.77 for the correlations of the exhumation with the phase velocity of the 65 s Rayleigh wave and with the LAB depth, respectively.

## Seismicity

Seismicity of Great Britain and Ireland was taken from the British Geological Survey (BGS) seismicity catalogue[93,94]. Earlier joint analysis of seismic catalogues and tomographic models at a global scale showed that the thick, cold cratons had the highest seismic velocities and the lowest seismicity, on average[95,96]. At regional scales, correlations of seismicity and upper-mantle attenuation—which depends, in

large part, on the lithospheric temperature and thickness—also suggested lithospheric controls on the localisation of intraplate seismicity[65,97]. In this study, seismicity is correlated with the thickness of the lithosphere, which determines its geotherm and mechanical strength, so that the relationship and the underlying physical mechanism can be isolated.

## Data availability

The waveform data from the networks and arrays in Britain and Ireland used in this work are publicly available from international data centres (http://service.iris.edu, last accessed July 2025; https://www.orfeus-eu.org, last accessed July 2025; http://geofon.gfz-potsdam.de, last accessed July 2025).

## Code availability

Computer codes replicating the results in this study can be obtained from the corresponding author upon request. The thermodynamic models can be reproduced using the software Litmod (e.g., https://www.juanafonso.com/software, last accessed July 2025). The computation of correlation coefficients and fits were performed using Matlab (https://www.mathworks.com, last accessed July 2025). The figures were created with the Generic Mapping Tools[98].

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

## Acknowledgements

This research was supported by the UK Natural Environment Research Council Grants NE/X000060/1 and NE/Y000218/1, the Science Foundation Ireland (SFI) Grant Number 16/IA/4598, co-funded by the Geological Survey of Ireland and the Marine Institute, and Project InnerSpace (https://projectinnerspace.org, last accessed July 2025) (R.B. and S.L.). This work has been done in the framework of the project 4D Dynamic Earth funded by ESA (4000140327/23/NL/SD) as part of EXPRO+ (R.B. and S.L.). D.C. acknowledges support from Research Ireland through research grants 13/RC/2092_P2 (Research Ireland Centre for Applied Geosciences, iCRAG). J.F. acknowledges support from Grants PID2020-114854GB-C22 and CNS2022-135621 funded by Ministry of Science and Innovation, Spain MCIN/AEI/10.13039/501100011033 and European Union Next Generation EU/PRTR.

## Author contributions

R.B.: conceptualisation, methodology, software, validation, formal analysis, investigation, resources, data curation, writing, editing, visualisation. S.L.: conceptualization, methodology, software, validation, formal analysis, investigation, resources, writing, editing, project administration, funding acquisition. D.C.: conceptualization, validation, resources, data curation, writing. Y.X.: methodology, software, validation. J.F. methodology, software, validation. T.M.: methodology, software, validation.

## Competing interests

The authors declare no competing interests.

## Consent to publication

We, the undersigned authors, hereby consent to publication. We confirm that we have contributed to the work submitted and agree to its publication. We certify that the manuscript is original, has not been previously published, and is not under consideration for publication elsewhere. All authors have reviewed the manuscript and consent to its publication. We agree to abide by the journal's policies and ethical guidelines regarding authorship, conflicts of interest, and data transparency.

## Additional information

**Peer review information** : *Nature Communications* thanks Aurélien Mordret, Morgan Jones and the other, anonymous, reviewer(s) for their contribution to the peer review of this work. A peer review file is available.

