## [Transparent Peer Review file · Nature Communications]

Volcanism and long-term seismicity controlled by plume-induced plate thinning

Corresponding Author: Dr Raffaele Bonadio

Version 0:

Reviewer comments:

Reviewer #1

(Remarks to the Author)

The manuscript "Volcanism and long-term seismicity controlled by plume-induced plate thinning" describes new tomographic results based on the inversion of the phase velocity dispersion curves of Rayleigh and Love waves to obtain thermo-mechanical structures of the lithosphere and upper asthenosphere below the British and Irish islands. The authors found that a thinned lithosphere correlates remarkably with the surface expressions of volcanism, exhumation rates and the current seismicity. The authors used a newly developed method to perform this thermo-mechanical inversion which allows a more in-depth understanding of the Earth structure, beyond seismic velocity observables.

I really enjoyed reading this manuscript and I am truly impressed by the potential of seismic thermography. I only have minor comments, mostly for my personal curiosity, but which might also interest some readers and I recommend a prompt publication after they are addressed.

1) I was wondering what are the effects of varying viscosity in your inversion? Is it taken into account? Is it neglected? If neglected, what could be the biases induced on the inversion results? Can you comment on that?

2) Your inversion fits the dispersion curves adequately so we can expect that the inverted temperature profiles are somehow robust. As a sanity check, did you compare heat-flow measurements at the surface (from the Global Heat-Flow database) with the heat-flow map you would obtain by propagating the heat from your temperature model to the surface (assuming some standard heat production in the mantle and the crust)? Do they compare at least qualitatively?

Reviewer #2

(Remarks to the Author)

This manuscript is motivated by an apparent paradox: if volcanism of the North Atlantic Igneous Province (NAIP) is due to the Iceland mantle plume, it should be tightly concentrated in space at a given time; yet observations show that simultaneous volcanism occurred over lateral distances of the order of 1000 km. This suggests that either multiple mantle upwellings occurred over a broad region, or that melt was able to flow long distances away from the Iceland plume along pathways (channels) where the lithosphere is thin.

The authors analyze new surface wave (Rayleigh and Love waves) seismic data from Ireland. The surface wave phase velocities are a proxy for the depth to the lithosphere-asthenosphere boundary (LAB). The authors find a striking correlation between the distribution of Paleocene uplift and magmatism on the one hand, and the variations of the LAB depth on the other. The results show that uplift and magmatism in Britain and Ireland 60 Myr ago were concentrated where the lithosphere is thin. The authors speculate that hot Iceland plume material spread laterally away from the hotspot in channels where the lithosphere is thin, although they acknowledge that there is no direct evidence for the existence of such channels between Iceland and Ireland.

At line 399 the manuscript suddenly shifts gears to consider how the seismicity distribution in Britain and Ireland is related to plume-induced plate thinning. The authors find that the seismicity is concentrated in the same thin-lithosphere regions as the Paleocene magmatism. These last 1.5 pages of the manuscript detract somewhat from the main conclusion, which rather awkwardly appears 'in the middle' of the manuscript, before the section on seismicity. In my opinion this section should be deleted so that the main conclusion appears more clearly.

The authors' results represent a significant advance in our understanding of magmatism in the NAIP, and their conclusion that lithospheric thickness controls magmatism has importance beyond the study area. In my opinion the work is sufficiently important to appear in Nature Communications. In addition, the analysis appears to be very carefully done (although I am not a seismologist, and cannot comment on the applicability or robustness of the methods used). Finally, the manuscript is very clearly and correctly written.

I recommend publication of the manuscript in essentially its present form, but with the short section on seismicity deleted.

Reviewer #3

(Remarks to the Author)

The manuscript "Volcanism and long-term seismicity controlled by plume-induced plate thinning" by R. Bonadio and co-authors presents an interesting and well-presented data set of seismic data and thermography to identify crustal thinning associated with the emplacement of the North Atlantic Igneous Province (NAIP) in the British and Irish sector (BIPIP). I am not an expert in geophysics, so I am not best placed to find potential issues in how the data was processed. I do have knowledge of the NAIP though, so that is where I will focus my suggestions and comments. I recommend minor revisions.

1. The emplacement of the NAIP (including the BIPIP) was more complex than a single uplift and exhumation event at ~60 Ma. Evidence from Faroe-Shetland basin just north of the BIPIP indicates at least three distinct transient uplift events coinciding roughly with the Danian-Selandian, Selandian-Thanelian, and Paleocene-Eocene (Thanelian-Ypresian) boundaries (Conway-Jones and White, 2022; Hartley et al., 2011; Shaw Champion et al., 2008). The Antrim lavas broadly coincide with the first of these uplift events, but the vast majority of the NAIP emplacement (by volume) and the final rifting that led to seafloor spreading is associated with the third and final uplift event (e.g. Storey et al., 2007; Wilkinson et al., 2017). That would suggest, albeit it circumspectly, that the lithospheric thinning measured here is a culmination of these transient uplifts. This may or may not have a significant impact on the data and interpretations presented here, but it is important to discuss a little as it could have important implications for the degree of thinning associated with each uplift event (e.g., if significant thinning is required to form the Antrim lavas, does that suggest the first uplift event was the most erosive to the base of the lithosphere?).

2. While the orientation of the magmatic centres in the BIPIP do not align with ancient terrane sutures, there is a broad correlation with the orientation of mid-ocean transforms in the present day northeast Atlantic. In a similar fashion to the Snæfellsjökull peninsula in Iceland today, the development of transform faults may have developed crustal weaknesses that encouraged lateral magma movement off-axis and into the BIPIP. Some recent papers have managed to date felsic volcanics at the very extremities of these off-axis provinces, and the ages seem to broadly align with the second and third uplift events (Gernigon et al., 2024; Lisica et al., 2025; Morris et al., 2024). There may be corroborating evidence for the propagation of a sub-lithospheric plume in the timing of these most distal volcanic centres for the mechanisms proposed here.

Minor comments:

The line numbers don't line up with the lines, making this part difficult...

Line 114: There are other estimates for the location of the Iceland hotspot at this time in addition to Steinberger et al., (2019), and indeed its variation in position across the three uplift events (~62–55 Ma), so a little more is needed here.

Line 360: Depending on who you talk to, the idea of constant continental lithosphere between Iceland and the BIPIP is somewhat contentious. It could be that at least part of it is overthickened oceanic lithosphere from previous Iceland hotspot activity. Could the lithospheric thinning associated with the formation of a rifted margin play any part in the modern-day observations?

Line 381: 54 Ma is a significant amount of time after ~60 (i.e. 62) Ma, suggest rewording.

References mentioned:

Conway-Jones, B.W., White, N., 2022. Paleogene buried landscapes and climatic aberrations triggered by mantle plume activity. *Earth and Planetary Science Letters* 593, 117644.

Gernigon, L., Knies, J., Schönenberger, J., Piraquive, A., van der Lelij, R., Huyskens, M.H., Planke, S., Berndt, C., Jones, M., Millett, J.M., Mohn, G., Alvarez Zarikian, C.A., 2024. Understanding volcanic margin evolution through the lens of Norway's youngest granite. *Terra Nova*, 36, 250–257.

Hartley, R.A., Roberts, G.G., White, N., Richardson, C., 2011. Transient convective uplift of an ancient buried landscape. *Nature Geoscience* 4, 562-565.

Lisica, K., Augland, L.E., Stevenson, J.A., Jerram, D.A., Beresford-Browne, A., Jones, M.T., 2024. High-precision U–Pb geochronology of the Lundy igneous complex: implications for North Atlantic volcanism and the far-field Paleocene–Eocene ash record. *Journal of the Geological Society*, 182 (1), jgs2023-140.

Morris, A.M., Lambart, S., Stearns, M.A., Bowman, J.R., Jones, M.T., Mohn, G., Andrews, G., Millett, J., Tegner, C., Chatterjee, S., Frieling, J., Guo, P., Jolley, D.W., Cunningham, E.H., Berndt, C., Planke, S., Alvarez Zarikian, C.A., Betlem, P., Brinkhuis, H., Christopoulou, M., Ferré, E., Filina, I.Y., Harper, D.T., Longman, J., Scherer, R.P., Varela, N., Xu, W., Yager, S.L., Agarwal, A., Clementi, V.J. 2024. Evidence for Low-Pressure Crustal Anatexis During the Northeast Atlantic Break-Up. *Geochemistry, Geophysics, Geosystems*, 25 (7), e2023GC011413.

Shaw Champion, M.E., White, N.J., Jones, S.M., Lovell, J.P.B., 2008. Quantifying transient mantle convective uplift: An example from the Faroe-Shetland basin. *Tectonics* 27, TC1002.

Storey, M., Duncan, R., Swisher III, C., 2007. Paleocene-Eocene Thermal Maximum and the opening of the Northeast Atlantic. *Science* 316, 587-589.

Wilkinson, C., Ganerød, M., Hendriks, B., Eide, E., 2017. Compilation and appraisal of geochronological data from the North Atlantic Igneous Province (NAIP), in: Péron-Pinvidic, G., Hopper, J.R., Stoker, M.S., Gaina, C., Doornenbal, J.C., Funck, T., Ártung, U.E. (Eds.), *The NE Atlantic Region: A Reappraisal of Crustal Structure, Tectonostratigraphy and Magmatic Evolution*. Geological Society, London, Special Publications.

Version 1:

Reviewer comments:

Reviewer #3

(Remarks to the Author)

The authors have addressed all previous comments I had made and I congratulate them on an impressive piece of work. I believe the manuscript is now ready for publication. Two small points that the authors can address if they wish:

Line 87: There may be some modelling evidence and timing of melt flare ups for past asthenospheric flow in the case of HALIP. See Heyn et al. (2024) <https://doi.org/10.1029/2023GC011380>

Line 392: The spreading between Greenland and Eurasia likely started before 54 Ma; that date is believed to be first oceanic crust formation at a mid-ocean ridge. There is fairly good evidence that rifting (and therefore the formation of a thin lithospheric channel at the proto-MAR) started around 56 Ma (see Storey et al., 2007).

Dear Dr. Laura Frahm,

We sincerely appreciate the time and effort that you and the reviewers have invested in evaluating our manuscript, "Volcanism and long-term seismicity controlled by plume-induced plate thinning" (NCOMMS-24-76918-T). We are grateful for the constructive feedback, which has helped us improve the clarity and scientific impact of our work. Below, we provide a detailed response to each of the reviewers' comments, with all changes made in the revised manuscript indicated.

Reviewer #1:

The manuscript "Volcanism and long-term seismicity controlled by plume-induced plate thinning" describes new tomographic results based on the inversion of the phase velocity dispersion curves of Rayleigh and Love waves to obtain thermo-mechanical structures of the lithosphere and upper asthenosphere below the British and Irish islands. The authors found that a thinned lithosphere correlates remarkably with the surface expressions of volcanism, exhumation rates and the current seismicity. The authors used a newly developed method to perform this thermo-mechanical inversion which allows a more in-depth understanding of the Earth structure, beyond seismic velocity observables.

I really enjoyed reading this manuscript and I am truly impressed by the potential of seismic thermography. I only have minor comments, mostly for my personal curiosity, but which might also interest some readers and I recommend a prompt publication after they are addressed.

1) I was wondering what are the effects of varying viscosity in your inversion? Is it taken into account? Is it neglected? If neglected, what could be the biases induced on the inversion results? Can you comment on that?

Thank you for this comment. In the modelling scheme used in this study, we do not consider viscosity. When predicting surface topography, we assume local isostasy, computed by integrating upper-mantle density under the assumption of the thermal steady state (as detailed in the "Seismic thermography" section of the Methods). Thus, the effects of viscosity, particularly as related to mantle flow or time-dependent deformation, do not enter the modelling. Including viscosity would be essential in the context of dynamic topography or transient thermal processes at the base of the lithosphere. However, there are no indications in tomography or our thermal models for vigorous mantle flow beneath the region, and topography is relatively flat. Our models are thus unlikely to be affected significantly by viscosity-sensitive dynamic processes. In future studies, however, our thermal model of the upper mantle can be used to estimate the viscosity of the lithosphere and underlying mantle and its effects on mantle deformation and flow.

2) Your inversion fits the dispersion curves adequately so we can expect that the inverted temperature profiles are somehow robust. As a sanity check, did you compare heat-flow measurements at the surface (from the Global Heat-Flow database) with the heat-flow map you would obtain by propagating the heat from your temperature model to the surface (assuming some standard heat production in the mantle and the crust)? Do they compare at least qualitatively?

Thank you for pointing out this omission. One of the standard model predictions is surface heat flow, based on the near surface temperature gradient and thermal conductivity. Comparison of our results (Fig. 1 of this document) with the Global Heat Flow Database (<https://www.ihfc-iugg.org/products/global-heat-flow-database>) shows general agreement but can go only this far due to the sparsity of the global heatflow database sampling. Our results agree also

with previous regional maps (Mather et al., 2018; Chambers et al., 2024), with 50-60 mW/m² values across most of Ireland, in particular (see Figs. 1 and 2 of this document).

We have now added a panel in the plot in “Extended Data Fig. 3” and “Extended Data Fig. 4” for predicted and observed heat flow at the locations represented at each model grid point.

Reviewer #2:

This manuscript is motivated by an apparent paradox: if volcanism of the North Atlantic Igneous Province (NAIP) is due to the Iceland mantle plume, it should be tightly concentrated in space at a given time; yet observations show that simultaneous volcanism occurred over lateral distances of the order of 1000 km. This suggests that either multiple mantle upwellings occurred over a broad region, or that melt was able to flow long distances away from the Iceland plume along pathways (channels) where the lithosphere is thin.

The authors analyze new surface wave (Rayleigh and Love waves) seismic data from Ireland. The surface wave phase velocities are a proxy for the the depth to the lithosphere-asthenosphere boundary (LAB). The authors find a striking correlation between the distribution of Paleocene uplift and magmatism on the one hand, and the variations of the LAB depth on the other. The results show that uplift and magmatism in Britain and Ireland 60 Myr ago were concentrated where the lithosphere is thin. The authors speculate that hot Iceland plume material spread laterally away from the hotspot in channels where they lithosphere is thin, although they acknowledge that there is no direct evidence for the existence of such channels between Iceland and Ireland.

At line 399 the manuscript suddenly shifts gears to consider how the seismicity distribution in Britain and Ireland is related to plume-induced plate thinning. The authors find that the seismicity is concentrated in the same thin-lithosphere regions as the Paleocene magmatism. These last 1.5 pages of the manuscript detract somewhat from the main conclusion, which rather awkwardly appears 'in the middle' of the manuscript, before the section on seismicity. In my opinion this section should be deleted so that the main conclusion appears more clearly.

The authors' results represent a significant advance in our understanding of magmatism in the NAIP, and their conclusion that lithospheric thickness controls magmatism has importance beyond the study area. In my opinion the work is sufficiently important to appear in Nature Communications. In addition, the analysis appears to be very carefully done (although I am not a seismologist, and cannot comment on the applicability or robustness of the methods used). Finally, the manuscript is very clearly and correctly written.

I recommend publication of the manuscript in essentially its present form, but with the short section on seismicity deleted.

Thank you for your the suggestions. A shift in focus when introducing seismicity was not our intention. Starting with the abstract, we tried to convey how the seismicity shows another facet of the spectacular interconnections in the dynamic Earth system, which is, we think, a good reason to keep it in the paper. Apart from the abstract (lines 36–55), we also mention seismicity in the end of the introductory section, which introduces the topic of seismicity early in the text (lines 134–143): “The previously unknown lateral variations of the lithospheric thickness display striking correlations with the distributions of uplift, volcanism and underplating at ~60 Ma and, also, with the distribution of seismicity at present.”

Reviewer #3:

The manuscript “Volcanism and long-term seismicity controlled by plume-induced plate thinning” by R. Bonadio and co-authors presents an interesting and well-presented data set of seismic data and thermography to identify crustal thinning associated with the emplacement of the North Atlantic Igneous Province (NAIP) in the British and Irish sector (BIPIP). I am not an expert in geophysics, so I am not best placed to find potential issues in how the data was processed. I do have knowledge of the NAIP though, so that is where I will focus my suggestions and comments. I recommend minor revisions.

1. The emplacement of the NAIP (including the BIPIP) was more complex than a single uplift and exhumation event at ~60 Ma. Evidence from Faroe-Shetland basin just north of the BIPIP indicates at least three distinct transient uplift events coinciding roughly with the Danian-Selandian, Selandian-Thonetian, and Paleocene-Eocene (Thonetian-Ypresian) boundaries (Conway-Jones and White, 2022; Hartley et al., 2011; Shaw Champion et al., 2008). The Antrim lavas broadly coincide with the first of these uplift events, but the vast majority of the NAIP emplacement (by volume) and the final rifting that led to seafloor spreading is associated with the third and final uplift event (e.g. Storey et al., 2007; Wilkinson et al., 2017). That would suggest, albeit it circumspectly, that the lithospheric thinning measured here is a culmination of these transient uplifts. This may or may not have a significant impact on the data and interpretations presented here, but it is important to discuss a little as it could have important implications for the degree of thinning associated with each uplift event (e.g., if significant thinning is required to form the Antrim lavas, does that suggest the first uplift event was the most erosive to the base of the lithosphere?).

Thank you for this helpful comment. We have expanded the text to include more detailed information about the distinct transient uplift events (lines 106–113). While we agree that these events cannot be individually resolved using our data and methods, and do not change our interpretation, we acknowledge the importance of including this information, as suggested. Specifically, we now detail (lines 106–113):

“The region experienced at least three distinct transient uplift events—around 62 Ma, 59 Ma, and 56 Ma. The Antrim Lava Group was emplaced during the first of these events, while the remainder of the NAIP emplacement and the final rifting that led to seafloor spreading are associated with the third and final uplift phase [27–30].”

2. While the orientation of the magmatic centres in the BIPIP do not align with ancient terrane sutures, there is a broad correlation with the orientation of mid-ocean transforms in the present day northeast Atlantic. In a similar fashion to the Snæfellsjökull peninsula in Iceland today, the development of transform faults may have developed crustal weaknesses that encouraged lateral magma movement off-axis and into the BIPIP. Some recent papers have managed to date felsic volcanics at the very extremities of these off-axis provinces, and the ages seem to broadly align with the second and third uplift events (Gernigon et al., 2024; Lisica et al., 2025; Morris et al., 2024). There may be corroborating evidence for the propagation of a sub-lithospheric plume in the timing of these most distal volcanic centres for the mechanisms proposed here.

Thank you for this helpful comment. Generally, we discuss the lateral transport of hot plume material in the sublithospheric mantle, which undergoes partial melting and gives rise to basaltic magmatism. The occurrence of felsic volcanoes at the far periphery of NAIP, however, is an

interesting observation and raises questions regarding their origin and possible relationship to the Iceland Plume. We have expanded the text to include more details as suggested (lines 344–350):

“Felsic magmatism of NAIP age has also been dated recently at the far reaches of the province, as far as southern Britain and just west of Norway, also hinting at a possible relationship with the Iceland Hotspot [51–53].”

Minor comments:

- *The line numbers don't line up with the lines, making this part difficult...*

Apologies for that, we just used the journal's LaTeX package. We can understand the comments clearly though and relate them to the text!

- *Line 114: There are other estimates for the location of the Iceland hotspot at this time in addition to Steinberger et al., (2019), and indeed its variation in position across the three uplift events (~62–55 Ma), so a little more is needed here.*

We have added additional references here and throughout the manuscript to ensure greater completeness.

- *Line 360: Depending on who you talk to, the idea of constant continental lithosphere between Iceland and the BIIP is somewhat contentious. It could be that at least part of it is overthickened oceanic lithosphere from previous Iceland hotspot activity. Could the lithospheric thinning associated with the formation of a rifted margin play any part in the modern-day observations?*

Thank you for your comment. While our dataset does not allow us to investigate the nature of the lithosphere between Iceland and the BIIP in detail, we acknowledge that this is a debated topic, particularly the distinction between stretched continental lithosphere and overthickened oceanic lithosphere linked to earlier Icelandic plume activity. We agree that lithospheric thinning related to rifted margin formation may play a role in modern-day geophysical signals, and a more detailed understanding would require further investigation.

- *Line 381: 54 Ma is a significant amount of time after ~60 (i.e. 62) Ma, suggest rewording.*

Thank you for pointing this out. We have modified the text for greater precision, following your suggestion.

Additional tracked changes to the first submitted manuscript

- Reference [43] was previously duplicated as [67] in the first submission document; this has now been corrected.

- The bibliography has been updated with additional references, which has resulted in changes to the numbering.

Final remarks

We are grateful for the reviewers' constructive feedback, which has helped us refine our manuscript. We believe that the revisions have strengthened our work, and we hope that it is now suitable for publication in Nature Communications. Please let us know if further clarifications if needed.

Sincerely,

Raffaele Bonadio, on behalf of all the authors.

Bibliography

Conway-Jones, B.W., White, N., 2022. Paleogene buried landscapes and climatic aberrations triggered by mantle plume activity. *Earth and Planetary Science Letters* 593, 117644.

Gernigon, L., Knies, J., Schönenberger, J., Piraquive, A., van der Lelij, R., Huyskens, M.H., Planke, S., Berndt, C., Jones, M., Millett, J.M., Mohn, G., Alvarez Zarikian, C.A., 2024. Understanding volcanic margin evolution through the lens of Norway's youngest granite. *Terra Nova*, 36, 250–257.

Hartley, R.A., Roberts, G.G., White, N., Richardson, C., 2011. Transient convective uplift of an ancient buried landscape. *Nature Geoscience* 4, 562-565.

Lisica, K., Augland, L.E., Stevenson, J.A., Jerram, D.A., Beresford-Browne, A., Jones, M.T., 2024. High-precision U–Pb geochronology of the Lundy igneous complex: implications for North Atlantic volcanism and the far-field Paleocene–Eocene ash record. *Journal of the Geological Society*, 182 (1), jgs2023-140.

Morris, A.M., Lambart, S., Stearns, M.A., Bowman, J.R., Jones, M.T., Mohn, G., Andrews, G., Millett, J., Tegner, C., Chatterjee, S., Frieling, J., Guo, P., Jolley, D.W., Cunningham, E.H., Berndt, C., Planke, S., Alvarez Zarikian, C.A., Betlem, P., Brinkhuis, H., Christopoulou, M., Ferré, E., Filina, I.Y., Harper, D.T., Longman, J., Scherer, R.P., Varela, N., Xu, W., Yager, S.L., Agarwal, A., Clementi, V.J. 2024. Evidence for Low-Pressure Crustal Anatexis During the Northeast Atlantic Break-Up. *Geochemistry, Geophysics, Geosystems*, 25 (7), e2023GC011413.

Shaw Champion, M.E., White, N.J., Jones, S.M., Lovell, J.P.B., 2008. Quantifying transient mantle convective uplift: An example from the Faroe-Shetland basin. *Tectonics* 27, TC1002.

Storey, M., Duncan, R., Swisher III, C., 2007. Paleocene-Eocene Thermal Maximum and the opening of the Northeast Atlantic. *Science* 316, 587-589.

Wilkinson, C., Ganerød, M., Hendriks, B., Eide, E., 2017. Compilation and appraisal of geochronological data from the North Atlantic Igneous Province (NAIP), in: Péron-Pinvidic, G., Hopper, J.R., Stoker, M.S., Gaina, C., Doornenbal, J.C., Funck, T., Ártung, U.E. (Eds.), *The NE Atlantic Region: A Reappraisal of Crustal Structure, Tectonostratigraphy and Magmatic Evolution*. Geological Society, London, Special Publications.

Mather, B. R., Farrell, T. F., & Fulla, J. (2018). Probabilistic surface heat flow estimates assimilating paleoclimate history: New implications for the thermochemical structure of Ireland. *Journal of Geophysical Research: Solid Earth*, 123, 10,951–10,967.
<https://doi.org/10.1029/2018JB016555>

Chambers, E., Fulla, J. Kiyan, D., Lebedev, S., Bean, C.J., Meere, P., Daly, J.S., Noller, N.W., Raine, R., Blake, S, O'Reilly, B.M. (2024). A new 3D temperature model for Ireland from joint geophysical-petrological inversion of seismic, surface heat flow and petrophysical data.
<https://eartharxiv.org/repository/view/7968/>

Figure 1: Predicted heat flow in the study region. Black contours indicate estimated magmatic underplating in km (Tomlinson et al., 2006)

Figure 2: Observed heat flow in the study region (dataset from Mather et al., 2018)

Dear Dr. Laura Frahm,

We truly appreciate the time and effort you and the reviewers have dedicated to evaluating our manuscript "Volcanism and long-term seismicity controlled by plume-induced plate thinning" (NCOMMS-24-76918B). We are thankful for the constructive feedback, which has helped us enhance the scientific impact of our work. Below, we address the comments from reviewer #3 in detail, with corresponding changes marked in the revised manuscript.

Reviewer #3:

The authors have addressed all previous comments I had made and I congratulate them on an impressive piece of work. I believe the manuscript is now ready for publication. Two small points that the authors can address if they wish:

- Line 87: There may be some modelling evidence and timing of melt flare ups for past asthenospheric flow in the case of HALIP. See Heyn et al. (2024) <https://doi.org/10.1029/2023GFC011380>

We agree that the volcanism in the region was not necessarily caused by a plume head arrival (e.g., lines 118–130). Heyn et al. (2024) modelling provides a plausible explanation for the distribution of volcanism in HALIP. At lines 85–89, we state that there is no direct evidence of asthenospheric flow in ancient LIPs, as the asthenospheric flow has changed since the time of magmatism, as we also mention. We agree, however, that the findings from Heyn et al. (2024) are relevant and worth mentioning, and we have now included this reference in the main text and the bibliography.

Line 392: The spreading between Greenland and Eurasia likely started before 54 Ma; that date is believed to be first oceanic crust formation at a mid-ocean ridge. There is fairly good evidence that rifting (and therefore the formation of a thin lithospheric channel at the proto-MAR) started around 56 Ma (see Storey et al., 2007).

Thank you for this comment. We have now corrected this in the revised manuscript. We now detail (lines 392–394):

“The spreading between Greenland and Eurasia starting at ~56 Ma [28] created a thin-lithosphere channel beneath the newly formed Mid-Atlantic Ridge.”

Additional tracked changes to the first submitted manuscript

- Longitude and latitude annotations have been added to Extended Data Fig. 2 as requested in the author checklist document.

- The bibliography has been updated with additional references in response to the first comment of reviewer #3, which has resulted in changes to the numbering.

- Bibliography reference

Xu, Y., Lebedev, S., Civiero, C., Fulla, J.: Global and tectonic-type physical reference models of the upper mantle. EGU General Assembly 2023, Vienna, Austria, 24-28 Apr 2023 (2023) <https://doi.org/10.5194/egusphere-egu23-5459>

has been updated to a most recent, just published work

Xu, Y., Lebedev, S., Fulla, J.: Average physical structure of cratonic lithosphere, from thermodynamic inversion of global surface-wave data. *Mineralogy and Petrology* (2025) <https://doi.org/10.1007/s00710-025-00926-0>

Final remarks

We hope that the revised manuscript meets the requirements for publication in Nature Communications. Please do not hesitate to let us know if any further clarification is needed.

Sincerely,

Raffaele Bonadio, on behalf of all the authors.

Bibliography

[20] Heyn, B. H., Shephard, G. E., & Conrad, C. P. Prolonged multi-phase magmatism due to plume-lithosphere interaction as applied to the High Arctic Large Igneous Province. *Geochemistry, Geophysics, Geosystems*, 25, e2023GC011380. (2024) <https://doi.org/10.1029/2023GC011380>

[28] Storey, M., Duncan, R., Swisher III, C.: Paleocene-Eocene Thermal Maximum and the Opening of the Northeast Atlantic. *Science* 316 (2007) <https://doi.org/10.1126/science.1135274>

Xu, Y., Lebedev, S., Civiero, C., Fullea, J.: Global and tectonic-type physical reference models of the upper mantle. EGU General Assembly 2023, Vienna, Austria, 24-28 Apr 2023 (2023) <https://doi.org/10.5194/egusphere-egu23-5459>

[68] Xu, Y., Lebedev, S., Fullea, J.: Average physical structure of cratonic lithosphere, from thermodynamic inversion of global surface-wave data. *Mineralogy and Petrology* (2025) <https://doi.org/10.1007/s00710-025-00926-0>